# Comparison of the Survival Outcomes of Minimally Invasive Surgery with Open Surgery in Patients with Uterine-Confined and Node-Negative Cervical Cancer: A Population-Based Study

**DOI:** 10.3390/cancers15102756

**Published:** 2023-05-14

**Authors:** Seiji Mabuchi, Tomoyuki Sasano, Naoko Komura, Michihide Maeda, Shinya Matsuzaki, Tsuyoshi Hisa, Shoji Kamiura, Toshitaka Morishima, Isao Miyashiro

**Affiliations:** 1Department of Gynecology, Osaka International Cancer Institute, Osaka 541-8567, Japan; michihide.maeda@oici.jp (M.M.); shinya.matsuzaki@oici.jp (S.M.); tsuyoshi.hisa@oici.jp (T.H.); kamiura-sh@oici.jp (S.K.); 2Department of Obstetrics and Gynecology, Osaka Saiseikai Nakatsu Hospital, Osaka 530-0012, Japan; sasano106@gmail.com; 3Department of Obstetrics and Gynecology, Kaizuka City Hospitl, Osaka 597-0015, Japan; naonaokomura@gmail.com; 4Cancer Control Center, Osaka International Cancer Institute, Osaka 541-8567, Japan; morishima.t@oici.jp (T.M.); miyashir@oici.jp (I.M.)

**Keywords:** cervical cancer, MIS, open surgery, survival

## Abstract

**Simple Summary:**

To investigate the efficacy of minimally invasive surgery (MIS) in the treatment of uterine-confined and node-negative cervical cancer in Japan, we conducted a population-based study using Osaka Cancer Registry data ranging between 2011 and 2018. A total of 2279 patients who underwent surgical treatment for uterine-confined and node-negative cervical cancer were identified. The patients were classified into groups according to surgery type (open and MIS groups) and year of diagnosis (2011–2014 and 2015–2018), and their oncologic outcomes were compared between the MIS and open groups. In the analyses including all patients, i.e., patients diagnosed in 2011–2014 and those diagnosed in 2015–2018, there were no differences in overall survival between the MIS and open groups. Our population-based cohort study provides epidemiological evidence that MIS does not compromise survival outcomes when compared with conventional open surgery in Japanese patients with FIGO 2018 stage I cervical cancer.

**Abstract:**

We aimed to compare the oncological outcomes between Japanese women with uterine-confined and node-negative cervical cancer who underwent open surgery and those who underwent minimally invasive surgery (MIS). A population-based retrospective cohort study was conducted using data from the Osaka Cancer Registry that ranged from 2011 to 2018. A total of 2279 patients who underwent surgical treatment for uterine-confined and node-negative cervical cancer were identified. The patients were classified into groups according to surgery type (open and MIS groups) and year of diagnosis (group one, 2011–2014; group two, 2015–2018). The oncologic outcomes were compared between the MIS and open groups. When the MIS group (n = 225) was compared with open group (n = 2054), overall, there was no significant between-group difference in terms of overall survival. Based on Kaplan–Meier estimates, the probability of overall survival at four years was 99.5% in the MIS group and 97.2% in the open group (*p* = 0.1110). When examined according to the year of diagnosis, there were no significant between-group differences in the overall survival in both groups one and two. In this population-based cohort study, MIS did not compromise survival outcomes when compared with conventional open surgery in Japanese patients with uterine-confined and node-negative (FIGO 2018 stage I) cervical cancer.

## 1. Introduction

Cervical cancer is the fourth most frequently diagnosed cancer in women, having an estimated incidence of 604,127 cases worldwide in 2020 [1]. 

Radical hysterectomy followed by tailored adjuvant radiotherapy has long been employed as a potentially curative treatment for early-stage cervical cancer [2,3]. Surgical options include traditional open surgeries or minimally invasive surgeries (MISs) (i.e., laparoscopic or robotic-assisted laparoscopic surgeries). Since the late 2000s, the pros and cons of the use of MISs to treat gynecological cancers have been intensively debated [4,5,6]. In the area of cervical cancer, data from a randomized controlled trial (the Laparoscopic Approach to Cervical Cancer [LACC]) that compared MIS and open surgery was published in 2018. The authors demonstrated a significantly increased risk of recurrence and decreased disease-free survival after MIS compared with open surgery [7]. These findings were confirmed by an epidemiological study using the National Cancer and Surveillance, Epidemiology, and End Results (SEER) program databases [8]. Based on these findings, treatment guidelines such as the National Comprehensive Cancer Network (NCCN) and European Society of Gynecological Oncology (ESGO) have changed their recommendations in favor of open surgery in the surgical management of early-stage cervical cancer [9,10]. However, a nationwide study conducted in the Netherlands did not find evidence of MIS being inferior to open surgery for treating early-stage cervical cancer [11], and the same findings were obtained by a retrospective study conducted in the USA [12]. Moreover, a recent meta-analysis suggested that laparo-assisted vaginal radical hysterectomies are safe and effective for early-stage cervical cancer [13]. Importantly, in the subgroup analysis of the LACC study [7], there was no survival difference between MIS and open surgery with respect to patients with small-volume disease (<2 cm), and a similar finding was observed in other retrospective studies [14,15]. Collectively, these results indicate that the conclusions of the abovementioned landmark studies could not be generalized for all women with cervical cancer receiving MIS; they may be affected by tumor size, the country in which the MIS was performed, or the surgical procedures employed.

In Japan, laparoscopic surgeries for gynecological malignancies increased by approximately 1.8-fold (from 1898 to 3490 cases) between 2014 and 2016 [16]. Robotic gynecological surgery has also rapidly increased in usage after being covered by the National Health Insurance in April 2018 [17]. However, these MIS procedures have been mainly employed in patients with endometrial cancer or benign conditions. In patients with invasive cervical cancer, only laparoscopic approach has been covered by the National Health Insurance in Japan. Moreover, in Japan, the oncologic safety of MIS in the treatment of cervical cancer has been evaluated only in several retrospective studies [18,19,20,21,22]. Therefore, the oncological safety of MISs over open surgery in Japanese women with early-stage cervical cancer remains unclear.

To investigate the efficacy of MISs in the treatment of uterine-confined and node-negative (International Federation of Gynecology and Obstetrics [FIGO] 2018 stage I) cervical cancer in Japan, we retrospectively compared the oncologic outcomes of open and MIS procedures using data obtained from the population-based cancer registry in Osaka Prefecture.

## 2. Materials and Methods

### 2.1. Data Source 

This retrospective, observational study was conducted in Osaka Prefecture, Japan, using data obtained from the population-based Osaka Cancer Registry (OCR). The OCR is a long-term survey that gathers data on the identification and management of all malignancies found in Osaka Prefecture and has been in operation since 1962 [23].

The OCR records all new cases of cancer that are discovered by reports from healthcare facilities or databases holding death certificates. The patient data that we obtained from the OCR included sex, cancer diagnosis age, date of diagnosis, date of death, and date of last follow-up of vital status. Tumor-specific data included the site of cancer, extent of disease, histology, and date of cancer diagnosis. The extent of disease was divided into the three groups: (1) localized: cancer only affects the original organ; (2) regional: regional lymph nodes and/or nearby tissues have been affected by cancer; and (3) distant: distant organs have been affected by cancer metastasis. The following is a correlation between the severity of the disease and its FIGO 2018 classification: localized, stage I; regional, stage II, III, and IVA; and distant metastasis, stage IVB. Using the morphological code from the International Classification of Diseases for Oncology, Third Edition (ICD-O3M), the histological subtype was determined. Treatment information included the type of initial treatment (surgery, chemotherapy, and radiation therapy). However, the OCR does not gather data on the socioeconomic status of patients, their history of comorbid conditions, their subsequent medical care, or their reasons of death. The follow-up of the vital status of cancer patients is routinely performed using death certificate [23].

### 2.2. Study Population 

The inclusion criteria were as follows: (1) uterine cervical neoplasia cases (International Classification of Disease, 10th revision Code: C53, malignant neoplasm of the cervix uteri) registered in the OCR between 2011 and 2018; (2) patients with three common histologic subtypes: squamous cell carcinoma, adenocarcinoma, and adenosquamous carcinomas; (3) patients aged 18–80 years; and (4) patients living in Osaka at the time of cervical cancer diagnosis who had received conventional open surgery (Open group) or laparoscopic or robotic-assisted surgeries (MIS group) for “localized” cervical cancer at medical facilities in Osaka. Patients having multiple malignancies or missing surgical type information and whose extent of disease was distant, regional, or unknown were not included in the analyses. Survival was analyzed in 2279 women with cervical cancer (Figure 1).

### 2.3. Statistical Analysis

Overall survival (OS) was defined as the period of time between the diagnosis of cervical cancer to the date of death or last follow-up visit. The patients were divided into two groups according to the year of diagnosis (Group one: early-phase, 2011–2014; Group two: late phase, 2015–2018). The OS was then compared between the open and MIS groups based on the Kaplan–Meier method, and the log-rank test was used to compare the outcomes. The Student’s *t*-test, Wilcoxon rank sum test, or median test were used where appropriate to compare continuous data between the groups. A chi-square test or Fisher’s two-tailed exact tests were used where appropriate to compare frequency counts and proportions between the groups. All analyses were conducted using Microsoft Office Excel 2019 and GraphPad Prism 9.5.1, and a *p*-value of <0.05 was considered statistically significant.

## 3. Results

### 3.1. Investigations Involving All Uterine Confined and Node-Negative Cervical Cancer Patients

Between 1 January 2011 and 31 December 2018, 6513 women in Osaka were diagnosed with cervical cancer. The study population selection is shown in Figure 1. A total of 2279 uterine-confined and node-negative (FIGO 2018 stage I) cervical cancer patients who were treated with either MIS (MIS group: n = 225) or open surgery (open group: n = 2054) were included in the analysis (Figure 1).

The clinicopathological characteristics of the patients in the two groups are presented side-by-side in Table 1. Open surgery was performed in most of the patients (90.1%). When the two groups were compared, patients in the MIS group were older than those who were in the open group. The proportion of patients with adenocarcinoma or adenosquamous carcinoma histology was greater in the MIS group than in the open group. The majority of patients (78.2% in the MIS group and 79.2% in the open group) were treated with surgery alone.

As shown in Figure 2, there was no significant between-group difference in the OS. Based on the Kaplan–Meier estimates, the probability of OS at four years was 99.5% in the MIS group and 97.2% in the open group (*p* = 0.1110).

### 3.2. Investigations According to Year of Diagnosis

To investigate the potential effect of the diagnosis year, we divided the patients into two groups according to the year of diagnosis: 2011–2014 (group one) and 2015–2018 (group two). The clinicopathologic characteristics of the patients in groups one and two are presented side-by-side in Table 2 according to the type of surgery (MIS vs. open surgery). As shown in the table, in group one (2011–2014), 3.9% of the patients were treated with MIS and 96.1% were treated with open surgery. In group two (2015–2018), 16.2% of the patients were treated with MIS and 83.8% were treated with open surgery. When group one was compared with group two, the proportion of patients treated with MIS was significantly increased (3.9% vs. 16.2%; *p* < 0.0001). As shown in the table, in group one, the are no significant differences in patient characteristics between the MIS group and open group. However, in group two, patients in the MIS group were older than those in the open group. Moreover, the proportion of patients with adenocarcinoma or adenosquamous carcinoma histology was greater in the MIS group than in the open group.

In the survival analyses, as shown in both groups one (Figure 3A) and two (Figure 3B), there were no significant differences in the OS according to the type of surgery. Based on the Kaplan–Meier estimates, the probability of OS at four years in group one was 100% in the MIS group and 96.3% in the open group (*p* = 0.1901). In group two, the estimated four-year OS rates were 99.4% and 98.4% in the MIS and open groups, respectively (*p* = 0.8008). 

## 4. Discussion

In this study, we compared the oncologic outcomes of open surgery and MIS for the treatment of uterine-confined and node-negative (FIGO 2018 stage I) cervical cancer using data from the population-based cancer registry in Osaka Prefecture. We found that MIS did not compromise survival outcomes when compared with conventional open surgery for uterine-confined and node-negative (FIGO 2018 stage I) cervical cancers. To the best of our knowledge, this population-based cohort study is among the largest in terms of comparing oncologic outcomes according to the mode of surgery (MIS vs. open surgery) in Japanese women with FIGO 2018 stage I cervical cancer.

In the current study, the observed survival rates in both the MIS group and open group were quite high (Figure 1). The precise reason for this remains unclear, as the information regarding tumor size, presence or absence of lymphovasucular space invasion, or extent of stromal invasion was not included in the OCR database. However, the fact that 78.2% of patients in the MIS group and 79.2% in the open group were treated with surgery alone (Table 1) indicates that the vast majority of patients included in the current study were at “low-risk” for postsurgical recurrence. According to previous reports, low-risk patients who do not exhibit any pathological risk factors have achieved postsurgical recurrence rates of 2–3% without requiring adjuvant treatment [3,24]. Moreover, our results obtained from FIGO 2018 stage I cervical cancer patients on survival outcome, which is a comparative oncologic outcome between the MIS and open groups, are consistent with those of a recent nationwide study conducted in the Netherlands [11] and a retrospective study conducted in the USA [12]. In both studies, early-stage cervical cancer patients were analyzed, and the disease-free survival and OS after MIS were comparable to the rates obtained after open surgery [11,12].

Although our results suggest the oncological safety of MIS in FIGO 2018 stage I cervical cancer patients, we do not recommend the use of MIS for cervical cancer patients without reflecting on the results of the laparoscopic approach to cervical cancer trial (LACC trial) [7], as the ESGO and NCCN have changed their recommendations in favor of open surgery in the surgical management of cervical cancer [9,10]. As suggested by the investigators of the LACC trial, to ensure the oncologic safety of MISs, efforts should be made to avoid the use of a vaginal uterine manipulator or intracorporeal colpotomy as they may increase the propensity for tumor dissemination [15,22,25,26]. Importantly, retrospective studies conducted after the LACC trial have suggested the possibility that MIS and open surgery can achieve comparable oncologic outcomes in patients with FIGO 2009 stage IB and tumor size of <2 cm [14,15,26]. Li et al. reported comparable oncologic outcomes between laparoscopic radical hysterectomy (LRH) and open radical hysterectomy (ORH) among patients with FIGO 2009 stage IB1 cervical cancer and tumor size of <2 cm in a multi-institutional retrospective cohort study that included 1484 patients (LRH, n = 585 vs. ORH, n = 899) [26]. Moreover, in their single-institutional retrospective cohort study, Kim et al. also reported that LRH did not influence disease recurrence in FIGO 2009 stage IB1 patients with cervical masses of ≤2 cm [14]. Similarly, in an international European retrospective cohort study (the SUCCOR study) in which patients with FIGO 2009 stage IB1 cervical cancer who had a tumor size of <4 cm were included, similar survival outcomes were observed between MIS radical hysterectomy and ORH in patients with a tumor size of ≤2 cm [15]. Consistent with these findings, the superiority of open surgery over MIS was not observed among patients with small cervical tumors (≤2 cm) in the abovementioned epidemiological study when using the National Cancer Database and SEER program database [8]. Therefore, although tumor size could not be examined in the current study, we believe that MIS can be employed as an effective treatment for uterine-confined and node-negative (FIGO 2018 stage I) cervical cancer patients.

Another important factor that should be contemplated when considering MIS for patients with cervical cancer is the surgeon’s proficiency in MIS. A recent retrospective study conducted in Japan suggested that a surgeon’s MIS proficiency (technical skills and abilities) was associated with oncological outcomes in patients with uterine cervical cancer treated with LRH [27]. Similar findings were observed in patients with colorectal cancer [28] or esophageal cancer [29]. In Japan, the number of laparoscopic surgeries for gynecological malignancy rapidly increased after 2014 [16]. Moreover, the number of laparoscopic radical hysterectomy for uterine cervical cancer also increased after 2014–2015 [15]. Therefore, in the current study, we divided the patients into two groups according to the year of diagnosis (early phase [group one]: diagnosed between 2011 and 2014; late phase [group two]: diagnosed between 2015 and 2018). However, we could not find any potential prognostic impact of the year of diagnosis (Figure 3A,B). The OS rates of the patients in the open group were comparable with those observed in the MIS group in both group one (early phase: diagnosed in 2011–2014) and group two (late phase: diagnosed in 2014–2018). However, we believe that MIS for cervical cancer should be performed by experienced surgeons, as it is impossible to exclude the possibility that the number of surgeries performed by the surgeon may impact the efficiency of MIS surgery.

Our analysis of the long-term cancer registry data is the strength of the present study. Since the OCR was founded, numerous initiatives have been taken to enhance the system, and the percentage of death certificate-only cases has dropped from the initial 17% to 7%. An estimated nine million people live in Osaka Prefecture, which accounts for almost one-third of Japan’s overall population. In addition to the fact that Osaka’s population is roughly equal to that of Sweden, as the OCR meets the quality of international standards, the OCR has been regarded as comparable to a national database [30,31].

This study has some limitations. Firstly, this study is of retrospective nature. As a result, as shown in Table 1 and Table 2, the characteristics of patients in the MIS group and open group were not well-balanced. Secondly, there was a relatively small number of patients treated with MIS as well as a short follow-up period in patients in group two (diagnosed between 2015 and 2018). Thirdly, our findings may have been skewed because we were unable to account for potential confounding factors, such as patient socioeconomic status, smoking status, pre-existing comorbidity, performance status, postoperative adjuvant treatments, and hospital characteristics such as surgeon volume and infrastructure, as they were not included in the OCR database. Fourthly, although it was speculated that the laparoscopic approach (rather than the robotic approach) was the predominant MIS in the current study based on the current national health insurance approval status, as the OCR database did not include details of the types of MIS (robotic or laparoscopic) performed, we could not evaluate the utility of robotic and laparoscopic surgeries separately. Moreover, the type of hysterectomy or lymphadenectomy performed and the use of a vaginal uterine manipulator and intracorporeal colpotomy could not be included as confounders because of the lack of information. Fifthly, due to the lack of cause-of-death data in the OCR database, we were unable to determine cancer-specific mortality. Moreover, because the OCR used the SEER summary stage for cancer registration and because the extent of tumor was only classified as a localized or regional disease, the utility of MISs could not be evaluated according to the FIGO staging or tumor size; thus, we cannot exclude the possibility that the MIS group in the current study included a greater number of FIGO stage IA and smaller number of FIGO stage IB diseases compared with the open group. Finally, the present study is not representative of the general population in Japan because we only used data from Osaka Prefecture, which is unique in terms of it being the second smallest prefecture with the third largest population. The results and consequences of this study, however, also apply to urban areas in other regions of Japan as well as in other nations.

## 5. Conclusions

Using data from Osaka’s large-scale, population-based cancer registry, we compared the oncologic outcomes of open and MIS surgeries for the treatment of uterine-confined and node-negative (FIGO 2018 stage I) cervical cancer. We found that MISs did not compromise survival outcomes when compared with conventional open surgery in Japanese patients with FIGO 2018 stage I cervical cancer. Our population-based cohort study provides epidemiological evidence that MIS can be an alternative to open surgery in this patient population in Japan.

## Figures and Tables

**Figure 1 cancers-15-02756-f001:**
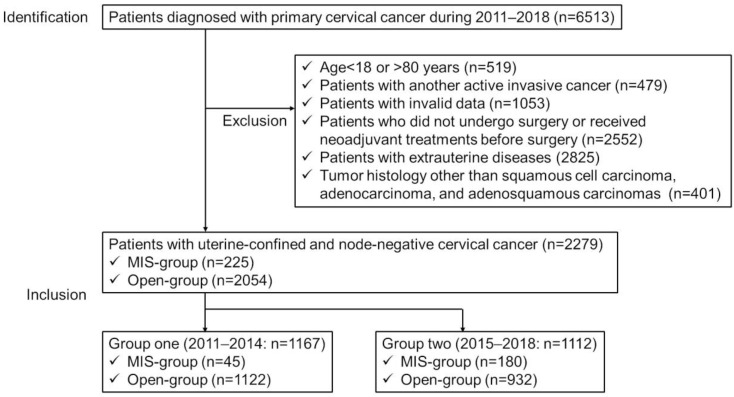
Flow diagram of the study population.

**Figure 2 cancers-15-02756-f002:**
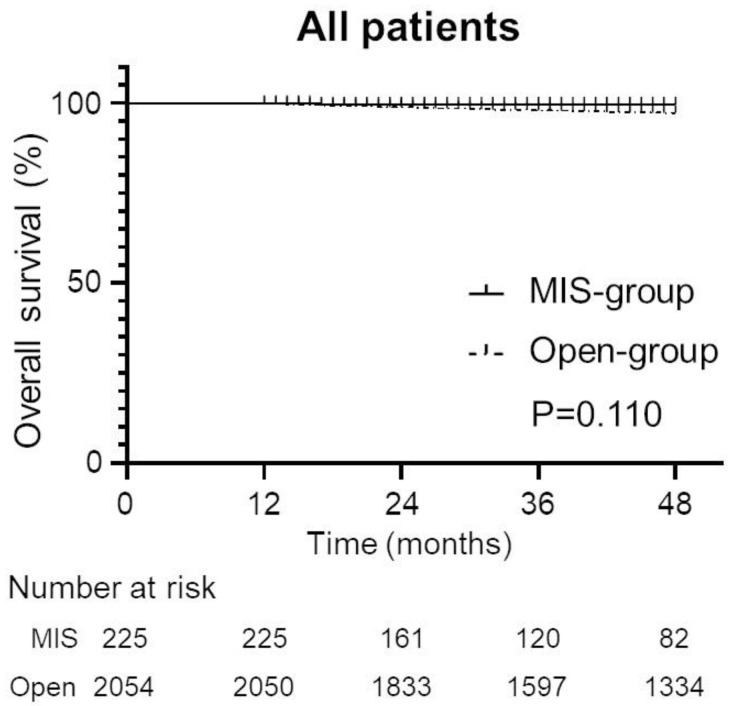
Kaplan–Meier estimates of the overall survival in uterine-confined and node-negative cervical cancer patients according to the type of surgery (MIS versus open surgery) for all patients (MIS group [n = 225] versus open group [n = 2054]; *p* = 0.1110).

**Figure 3 cancers-15-02756-f003:**
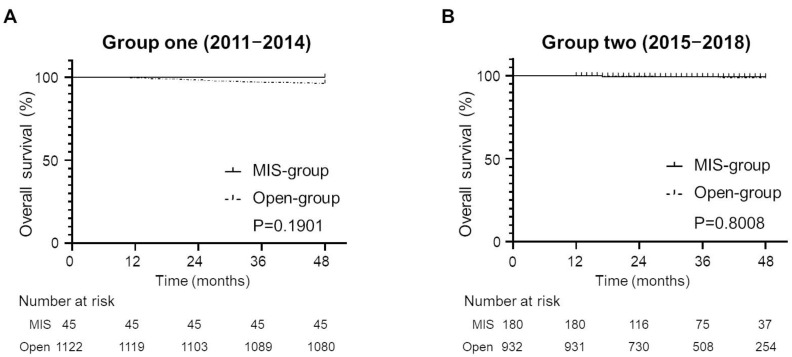
Kaplan–Meier estimates of overall survival in uterine-confined and node-negative cervical cancer patients according to the year of diagnosis (group one, diagnosed between 2011 and 2014; group two, diagnosed between 2014 and 2018; type of surgery (MIS versus open surgery). (**A**) Patients in group one (MIS group (n = 45) versus open group [n = 1122]; *p* = 0.1901). (**B**) Patients in group two (MIS group [n = 180] versus open group [n = 932]; *p* = 0.8008).

**Table 1 cancers-15-02756-t001:** Clinicopathological characteristics of all localized cervical cancer patients according to the type of surgery.

		MIS Group (n = 225) No. (%)	Open Group (n = 2054) No. (%)	*p*-Value
Age (years)	18–39	50 (22.2)	758 (36.9)	<0.0001
	40–60	133 (59.1)	966 (47.0)	
	61–80	42 (18.7)	330 (16.1)	
	Means ± SD	48.0 ± 11.6	45.5 ± 12.5	0.0024
Histological type	Squamous cell carcinoma	139 (61.8)	1497 (72.9)	0.0018
	Adenocarcinoma	76 (33.8)	482 (23.5)	
	Adenosquamous carcinoma	10 (4.4)	75 (3.6)	
Treatments	Surgery	176 (78.2)	1633 (79.5)	0.0016
	Surgery + Radiotherapy	5 (2.2)	102 (5.0)	
	Surgery + Chemotherapy	17 (7.6)	196 (9.5)	
	Surgery + Radiotherapy + Chemotherapy	27 (12.0)	123 (6.0)	

MIS, minimally invasive surgery; open, open surgery; SD, standard deviation.

**Table 2 cancers-15-02756-t002:** Clinicopathological characteristics of localized cervical cancer patients according to the year of diagnosis and type of surgery.

		Group One (2011–2014)			Group Two (2015–2018)		
		MIS Group (n = 45)	Open Group (n = 1122)	*p*-Value	MIS Group (n = 180)	Open Group (n = 932)	*p*-Value
Age (years)	18–39	10 (22.2)	394 (35.1)	0.0569	40 (22.2)	364 (39.1)	<0.0001
	40–60	30 (66.7)	544 (48.5)		103 (57.2)	422 (45.3)	
	61–80	5 (11.1)	184 (16.4)		37 (20.6)	146 (15.7)	
	Means ± SD	45.9 ± 10.3	45.6 ± 12.4	0.8867	48.8 ± 11.8	45.4 ± 12.6	0.0019
Histological type	Squamous cell carcinoma	27 (60.0)	825 (73.5)	0.0649	112 (62.2)	672 (72.1)	0.0290
	Adenocarcinoma	17 (37.8)	256 (22.8)		59 (32.8)	226 (24.2)	
	Adenosquamous carcinoma	1 (2.2)	41 (3.7)		9 (5.0)	34 (3.6)	
Treatments	Surgery	37 (82.2)	880 (78.4)	0.4064	139 (77.2)	753 (80.8)	0.0026
	Surgery + Radiotherapy	0 (0)	59 (5.3)		5 (2.8)	43 (4.6)	
	Surgery + Chemotherapy	6 (13.3)	118 (10.5)		11 (6.1)	78 (8.4)	
	Surgery + Radiotherapy + Chemotherapy	2 (4.4)	65 (5.8)		25 (13.9)	58 (6.2)	

MIS, minimally invasive surgery; open, open surgery; SD, standard deviation.

## Data Availability

The data that support the findings of this study are available from the corresponding author upon reasonable request.

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
