# Peer review of "Comparison of the Survival Outcomes of Minimally Invasive Surgery with Open Surgery in Patients with Uterine-Confined and Node-Negative Cervical Cancer: A Population-Based Study"

_cancers, 2023, doi:10.3390/cancers15102756_

Round 1

Reviewer 1 Report

This manuscript is significant which provides insight into the oncologic outcome of MIS for uterine-confined cervical cancer after the publication of the LACC study. However, this is a retrospective study, not an RCT, and many issues remain.

 Specific critics are the following:

1)    It is unclear why the subjects were divided by 2011-2014 and 2015-2018. A detailed description of the reason for dividing the cases into two groups at this point would be appreciated. 

2)    According to table 1, there are significant differences in patient characteristics between the MIS group and the Open-group in terms of age, histology, and treatments. Is it reasonable to compare the oncologic outcomes in these two groups? Detailed explanation is needed.

3)    As for the Stage I cervical cancer compared between MIS group and Open-group, is it possible that the MIS group has more Stage IA and the Open group has more Stage IB cases? The presence or absence of lymph node dissection is also unclear, and a detailed explanation of the possible confounding factor of stage bias is required.

4)    The latter part of the "2.2. study population" paragraph in "Materials and Methods" appears to be a copy of the submission instruction.

Author Response

Responses to Reviewer 1

We thank the reviewer for carefully reviewing our manuscript and providing us with useful comments. Our manuscript was revised based on the reviewer’s comments.

Comment 1:

It is unclear why the subjects were divided by 2011-2014 and 2015-2018. A detailed description of the reason for dividing the cases into two groups at this point would be appreciated. 

Response:

Thank you for the reviewer’s thoughtful comment. We have included a short discussion on the reason for dividing the cases into two groups in lines 253-255 of the revised manuscript.

Comment 2:

According to table 1, there are significant differences in patient characteristics between the MIS group and the Open-group in terms of age, histology, and treatments. Is it reasonable to compare the oncologic outcomes in these two groups? Detailed explanation is needed.

Response:

We totally agree with the reviewer’s comment. We have included a discussion on this issue as a limitation of the current study in lines 272-274 of the revised manuscript.

Comment 3:

As for the Stage I cervical cancer compared between MIS group and Open-group, is it possible that the MIS group has more Stage IA and the Open group has more Stage IB cases? The presence or absence of lymph node dissection is also unclear, and a detailed explanation of the possible confounding factor of stage bias is required.

Response:

As the reviewer pointed, we cannot exclude the possibility that MIS-group included greater number of FIGO stage IA and smaller number of FIGO stage IB diseases compared with Open-group. We have stated this as a limitation of this study in lines 291-293 of the revised manuscript.

Comment 4:

The latter part of the "2.2. study population" paragraph in "Materials and Methods" appears to be a copy of the submission instruction.

Response:

I have corrected the errors.

Reviewer 2 Report

Seiji Mabuchi and co-workers reported the title of Comparison of minimally invasive surgery with open approach in patients with uterine-confined and node-negative cervical cancer: a population-based study, which the experimental is carefully conducted, and the results have been presented correctly, and the contents fall well into the scope of the journal. I recommend the publication of this paper after minor revision :

1. The format of the figures (figures 1-3) should match the requirements of the magazine.

2. Please cite the following literatures : Micropor. Mesopor. Mater., 2022, 112098; and Inorganics, 2022, 10, 202.

Author Response

Responses to Reviewer 2

We thank the reviewer for carefully reviewing our manuscript and providing us with useful comments. Our manuscript was revised based on the reviewer’s comments.

Comment 1. 

The format of the figures (figures 1-3) should match the requirements of the magazine.

Response:

As suggested, we have revised the figures.

Comment 2. 

Please cite the following literatures : “Micropor. Mesopor. Mater., 2022, 112098; and “Inorganics, 2022, 10, 202”.

Response

I have tried to cite the literatures. However, they are the papers of different areas. So, we could not do it. However, as we received a comment from the editors to increase the number of references up to 30, we have included some more references in the revised manuscript.

Reviewer 3 Report

In this study, Seiji Mabuchi et al did a comparative population-based study on minimally invasive surgery (MIS) with open approach in patients with FIGO stage I cervical cancer. They found that MIS did not compromise survival outcomes when compared with conventional open surgery. The research was properly designed. A large population of patients were involved in the study, and the findings showed certain novelty and significance. However, there are some flaws and issues that need further clarification.

1.     The manuscript is not well written, with a lot of typographical and grammar mistakes. The whole flow is really confused and hard to follow. A thoroughly improvement in language editing is needed.

2.     The title is “Comparison of minimally invasive surgery with open approach in patients with uterine-confined and node-negative cervical cancer: a population-based study”. However, the major evaluation is patient survival. More analysis should be considered for the current title, such as disease free survival, recurrence rate, profile of complications, and etc.  

3.     In result 3.2, the results in table 2 is not was not adequately explained. The percentage of patients who treated with MIS was not clearly presented in table 2.

4.     In figure 1, it includes the expression of “Group one” and “Group 2”. Please revise in a consistent style.

The manuscript is not well written, with a lot of typographical and grammar mistakes. The whole flow is really confused and hard to follow. A thoroughly improvement in language editing is needed.

Author Response

Responses to Reviewer 3

We thank the reviewer for carefully reviewing our manuscript and providing us with useful comments. Our manuscript was revised based on the reviewer’s comments.

Comment 1.    

The manuscript is not well written, with a lot of typographical and grammar mistakes. The whole flow is really confused and hard to follow. A thoroughly improvement in language editing is needed.

Response

As suggested, we have corrected the typo and grammar mistakes as much as possible. The flow diagram has also been revised (Figure 1).

Comment 2.    

The title is “Comparison of minimally invasive surgery with open approach in patients with uterine-confined and node-negative cervical cancer: a population-based study”. However, the major evaluation is patient survival. More analysis should be considered for the current title, such as disease free survival, recurrence rate, profile of complications, and etc.  

Response

Thank you for the reviewer’s thoughtful comment. As disease free survival, recurrence rate or profile of complications cannot be evaluated in the present study due to the lack of information in OCR database, we have changed the title in the revised manuscript.

Comment 3.    

In result 3.2, the results in table 2 is not was not adequately explained. The percentage of patients who treated with MIS was not clearly presented in table 2.

Response

As suggested, we have revised the manuscript (lines 173-182 of the revised manuscript).

Comment 4.    

In figure 1, it includes the expression of “Group one” and “Group 2”. Please revise in a consistent style.

Response

As suggested, we have revised the Figure 1.

Round 2

Reviewer 1 Report

I express respect for the efforts of the authors who seriously responded to the strict question from the reviewer.I think this paper has been improved to increase its scientific value.

Reviewer 3 Report

The manuscript has been sufficiently improved.